# Effect of vascular simulation training on practice performance in residents: a retrospective cohort study

Lin Yang ,[1] Yanzi Li,[2] Jianlin Liu,[1] Yamin Liu[3]

LY and YL are joint first authors.

¹Vascular Surgery, First Affiliated Hospital of Xi'an Jiaotong University, Xi'an, China
²Department of Medical Administration, First Affiliated Hospital of Xi'an Jiaotong University, Xi'an, China
³International Radiology, First Affiliated Hospital of Xi'an Jiaotong University, Xi'an, China

**Correspondence to**
Dr Lin Yang; jdvascs@163.com

## ABSTRACT

**Objective** This study aims to investigate the teaching effect of vascular simulation training (ST) in rotating vascular residents.

**Design** Retrospective cohort study.

**Setting and participants** A total of 95 vascular residents were included from 2015 to 2018 in a university affiliated centre western China, and divided into an ST group and a conventional training (CT) group. The ST group received ST and CT, and the CT group only received CT.

**Primary outcome measures** Theoretical scores were assessed, and the technique parameters, complications and radiation damage of the procedures were analysed.

**Results** The mean scores (8.74±1.09 vs 8.13±1.31) and the rate of willingness for retraining (93.62% vs 79.17%) in residents were higher in the ST group than in the conventional training (CT) group (p<0.05). The success rate of arterial puncture was significantly higher in the ST group (78.72% vs 58.33%, p=0.03); however, the incidence of complications was similar between the two groups (p>0.05). The time of the puncture procedure was significantly lower (9.56±5.24 vs 12.15±6.87 min), and the comfort score of the patient (5.49±1.72 vs 4.71±1.57) was higher in the ST group than in the CT group (p<0.05). At the end of the assessment, the learning time for angiography (3.65±0.64 vs 4.07±0.77 months) and the complete procedure time (33.81±10.11 vs 41.32±12.56 min) were lower in the ST group than in the CT group (p<0.01). The fluo time for angiography (489.33±237.13 vs 631.47±243.65 s) and the cumulative air kerma (401.30±149.06 vs 461.16±134.14 mGy) were significantly decreased in ST group (p<0.05).

**Conclusion** The application of a vascular simulation system can significantly improve the clinical performance of residents and reduce the radiation damage from a single intervention procedure in patients.

## Strengths and limitations of this study

► The simulation training (ST) could promote the mean scores of residents.
► The ST could improve the clinical performance of residents.
► The ST could reduce the radiation damage.
► The ST should be wildly used in clinical teaching practice.
► The conclusions of this study should be confirmed via prospective randomised study.

mainly carried out through conventional teaching (CT) modes (including classroom teaching, lectures and surgical practice), and residents lack sufficient practical procedures with simulation training (ST) from theoretical knowledge to practical procedures, therefore, the true teaching effect was not ideal.

Three-dimensional vascular simulator systems (ANGIO Mentor system, Simbionix, Cleveland, Ohio, USA) use digital simulation to quantify the vascular interventional procedures of the cardiovascular, peripheral and cerebrovascular systems. Residents can use the system to select cases for ST; ultimately, the ST results are scored according to the operating steps of the system. This ST can promote the mastery of vascular procedure performance in residents,[4–6] and this system provides an opportunity to perform endovascular procedures in a safe environment as an educational tool for novice residents.

However, due to the late entry of simulation systems in China, there has been no report of the use of three-dimensional vascular simulation systems in clinical practice teaching in the area of vascular surgery. Therefore, we have retrospectively collected the teaching data of residents who received the ST and those who did not in our hospital. The aim of this study was to assess the effect of simulation-based training on improving the technical performance and subsequent clinical procedures of residents in vascular surgery.

## INTRODUCTION

In recent years, with changes in the disease spectrum of Chinese patients, the incidence of peripheral arterial disease has increased significantly, which has also caused a severe economic and social burden.[1] Therefore, it is becoming increasingly important to strengthen general and specialised vascular disease skills education in the training of medical residents.[2 3] In the past decade, the practical skills training of resident has been

## METHODS

### Study procedures

A total of 95 vascular resident trainees at the First Affiliated Hospital of Xi'an Jiaotong University were respectively collected in this study from Jan 2015 to Dec 2018, 47 vascular residents received simulation-based vascular training (ST group) for 2 weeks, and then they completed the last clinical training. The other 48 residents only completed conventional clinical training (CT group, including classroom teaching, lectures and surgical practice) without the simulation course, and all residents needed to complete 6 months of endovascular training in vascular surgery. A survey was administered to determine the demographics, academic degree, specialty level and previous work experiences that may have been relevant in terms of the residents' ability to learn vascular interventional skills.

Before the course was performed, the residents in the ST group received a standardised introduction to the endovascular simulator and performed a cerebrovascular angiography procedure. The 2-week curriculum consisted of theory teaching and 30–60 min lectures per day covering basic catheter-based interventions, cerebrovascular disease, superficial femoral artery disease and renal artery disease. The residents received 1-hour mentored simulator sessions per day and practiced carotid, renal and superficial femoral artery interventions. This course was performed by a tutor. Then, residents needed to complete the primary endovascular procedure with direct instruction. During the entire training process, each resident was required to complete the ST for no less than 1 hour per day. The course concluded with a final cerebrovascular angiography procedure performed on the simulator. The residents in the CT group underwent the basic 2-week curriculum consisting of theory teaching and 30–60 min lectures per day covering basic endovascular intervention procedures but without the ST course.

### Simulation system

The vascular simulator system (ANGIO Mentor system, Simbionix, Cleveland, Ohio, USA, figure 1) was composed of a standard desktop computer with software

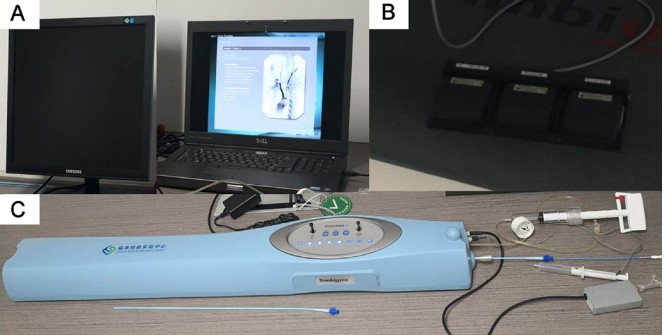

**Figure 1** The vascular simulator system (ANGIO Mentor system, Simbionix, Cleveland, Ohio, USA) used in this study simulated the vascular interventional procedures: (A) work station; (B) pedal; (C) simulator.

that simulated the human arterial system in three dimensions, and any user could perform the endovascular interventional procedures under the instruction of the system. This simulation system was connected to a haptic pressure feedback module, which used a force feedback system to detect external devices. When the users inserted the angiography catheters and wires, injected contrast and performed the endovascular procedures with digital subtraction angiography, all procedures could then be displayed on the screen in real time. The user was able to select the device to be simulated through one monitor, and the second monitor was used to display the simulated fluoroscopic image.

### Procedure evaluation

When the residents completed the training course, all residents received assessments of theoretical knowledge at 1 month and practical assessments at 6 months. The teaching expert group made the theoretical test questions based on the key points of vascular disease as well as the endovascular interventional procedures and technical points involved in vascular surgery clinical training (the theoretical knowledge test is formulated according to the standardised question bank and the simulation system question bank), and then the residents completed the assessment independently. Finally, the expert group determined the Student's score based on the results of the test (including the detailed information of knowledge and practice). The theoretical score range was 0–10, and a passing score was defined as a score higher than 7. All residents completed the arterial puncture procedure with the Seldinger technique and performed the cerebrovascular angiography procedure. The success criterion of arterial puncture was defined as follows: all puncture procedures were successfully completed, and the sheath was successfully inserted into the femoral artery. If the residents failed to complete the processes of puncture and sheath placement, which was defined as a puncture failure, then the puncture was performed by the teacher. The puncture success rate, puncture time and complications were recorded, and the comfort scores of the patients during the puncture procedure were assessed with a Numerical Rating Scale (NRS). The NRS Score ranged from 0 to 10, where 0 indicates worst uncomfortable pain and 10 indicates comfortable pain.[7] Subsequently, the residents needed to complete the cerebrovascular angiography procedure. The evaluation indicators of angiography were as following: learn time to complete the procedure (LTP, defined as the time from the beginning of training to the completion of the first angiography independently), time of complete procedure at the final test (TCP), fluo time of procedure (FTP), cumulative air kerma (CAK) and dose area product (DAP). TCP, FTP, CAK and DAP were defined as the time of procedure and related parameters of the assessed angiography procedure at the end of the training.

### Patient and public involvement

Patients and the public were not involved in the design or planning of the study.

| Table 1 | The baseline data of both groups | | |
| --- | --- | --- | --- |
| | CT group (n=48) | ST group (n=47) | P value |
| Sex (M) | 44 (91.67) | 42 (89.36) | 0.70 |
| Age (years) | 33.13±3.04 | 33.91±4.94 | 0.35 |
| Academic degree | | | 0.36 |
| Bachelor | 22 (45.83) | 26 (55.32) | |
| Postgraduate | 26 (54.17) | 21 (44.68) | |
| Specialty background | | | 0.48 |
| Vascular | 30 (62.50) | 26 (55.32) | |
| Non-vascular | 18 (37.50) | 21 (44.68) | |
| Work experience | | | 0.35 |
| >3 years | 16 (33.33) | 20 (42.55) | |
| ≤3 years | 32 (66.67) | 27 (57.45) | |

P value, compared with the CT group.
CT, conventional training; M, male; ST, simulation training.

| Table 2 | The performance of arterial puncture in residents | | |
| --- | --- | --- | --- |
| | CT group (n=48) | ST group (n=47) | P value |
| Puncture success rate | 28 (58.33) | 37 (78.72) | 0.033 |
| Total puncture success rate | 44 (91.67) | 45 (95.74) | 0.69 |
| Complications from puncture | 9 (18.75) | 8 (17.02) | 0.83 |
| Bruising | 5 (10.42) | 6 (12.77) | |
| Haematoma | 3 (6.25) | 2 (4.26) | |
| Infection | 0 | 0 | |
| Arteriovenous fistula | 0 | 0 | |
| Pseudoaneurysm | 1 (2.08) | 0 | |
| Time of puncture (min) | 12.15±6.87 | 9.56±5.24 | 0.002 |
| Comfort score of patients | 4.71±1.57 | 5.49±1.72 | 0.023 |

P value, compared with the CT group.
CT, conventional training; ST, simulation training.

## Statistical analysis

All data were analysed using SPSS V.11.0 (SPSS, Chicago, Illinois, USA), and p<0.05 was considered statistically significant. To test the difference between groups, we used $X^2$ analysis for categorical variables and Student's t-tests for continuous variables, and we tested the significance of the difference between two independent proportions when the results were presented as percentages.

## RESULTS

### The baseline data of the two groups

This study included 48 residents and 47 residents who were retrospectively recruited in this study, and there was no significant difference in baseline data between the two groups (table 1, p>0.05). There were 44 men and 4 women who received CT training, with a mean age of 33.13 years, and 42 men and 5 women who received ST training, with a mean age of 33.91 years (p>0.05). The background academic degrees were bachelor and postgraduate degrees in the CT and ST groups (p>0.05), and 30 and 26 residents had a specialty background in vascular surgery in both groups (62.50% vs 55.32%, p>0.05). The clinical work experience of most residents was less than 3 years in both groups (66.67% vs 57.45, p>0.05).

### The theoretical scores between both groups

All residents passed the training test; however, the mean scores of the residents were higher in the ST group than in the CT group (8.74±1.09 vs 8.13±1.31, p=0.014). After the clinical training, the training satisfaction rates of all residents were similar (97.87% vs 91.67%, p>0.05); however, when asked whether they wished to participate in the training again, the residents in the ST group showed a higher willingness rate than residents in the CT groups (93.62% vs 79.17%, p=0.04).

### The performance of arterial puncture in residents

After the training, all residents underwent an arterial puncture test (table 2), and the success rate of the procedure was higher in the ST group than in the CT group (78.72% vs 58.33%, p=0.033); however, the total puncture success rate was similar between the two groups (95.74% vs 91.67%, p>0.05). The complications of the puncture sites included bruising, haematoma, infection and pseudoaneurysm, and there was no significant difference in the incidence of complications between the ST and CT groups (17.02% vs 18.75%, p>0.05); however, the time of the puncture procedure was shorter in the ST group than in the CT group (9.56±5.24 vs 12.15±6.87 min, p=0.002), and the comfort score of patients was higher in the ST group than in the CT group according to the NRS Scores (5.49±1.72 vs 4.71±1.57, p=0.023).

### The outcome of cerebrovascular angiography

All residents needed to complete the final cerebrovascular angiography test; the related parameters are listed in table 3. The study curve of the residents in the ST group showed a lower mean LTP than residents in the CT groups (3.65 vs 4.07 months, p<0.01), and the TCP of the final test was higher in the CT group than in the ST group (41.32 vs 33.81 min, p=0.002). The radiation damage-related parameters were recorded, and the residents in the CT group showed higher mean values of FTP (631.47 vs 489.33 s, p<0.001) and CAK than the residents in the ST group (463.16 vs 401.30 mGy, p=0.043); however, the mean DAP value in both groups indicated no difference (128 624.30 vs 128 012.10 mGy.cm², p>0.052).

## DISCUSSION

The development of vascular surgery occurred relatively late in China; thus, the training model used for vascular residents in most university hospitals was traditional

**Table 3** The performance on cerebrovascular angiography in residents

| | CT group (n=48) | ST group (n=47) | P value |
|---|---|---|---|
| LTP (mon) | 4.07±0.77 | 3.65±0.64 | <0.01 |
| TCP (min) | 41.32±12.56 | 33.81±10.11 | 0.002 |
| FTP (s) | 631.47±243.65 | 489.33±237.13 | 0.005 |
| CAK (mGy) | 463.16±134.14 | 401.30±149.06 | 0.043 |
| DAP (mGy.cm$^2$) | 128 624.30±28 982.22 | 128 012.10±31 035.08 | 0.92 |

P value, compared with the CT group.

CAK, cumulative air kerma; CT, conventional training; DAP, dose area product; FTP, fluo time of procedure; LTP, learn time to complete procedure from beginning; ST, simulation training; TCP, time of complete procedure at the final test.

training; however, traditional training did not improve residents' understanding and interest in vascular surgery. In past decades, there were fewer residents who chose vascular surgery as a career option in China, which was consistent with the reports of previous studies.[7] Therefore, improving residents' interest in vascular surgery and promoting vascular clinical performance have been the main problems associated with clinical training.[8] Earlier studies have shown that compared with traditional training, the use of network media, social media and simulation systems can achieve better training results.[9–11] In this study, we used the vascular simulation system to assist residents in clinical training. The results revealed that ST could significantly improve the clinical practice effect of residents. The residents who received the ST had significantly higher theoretical scores; in addition, the interest level of residents was higher after ST. The vascular simulation system could standardise complex vascular systems and different vascular lesions through analogue calculations, which could help residents practice vascular skills training earlier and improve the interest and performance of residents.[3] Our result reveals that the residents could deeply understand theoretical knowledge and practical knowledge through ST, which also helps to improve the residents' theoretical and practical knowledge.

After ST and basic training, residents need further vascular operation training under the guidance of tutors, and the basic procedure for vascular residents is arterial puncture. However, it is impossible for residents to repeatedly perform the procedure during the treatment of patients; thus, in vitro ST has become the main teaching method in the training of vascular residents. Residents who received simulator training showed better clinical performance in the vascular surgery rotation, more motivation to learn, a shorter learning time and a lower number of clinical procedural errors, and the patients indicated a lower discomfort rate with the procedure.[12 13] In our study, the results confirmed that the residents who underwent the ST had a significantly higher success rate of arterial puncture, and the time of the puncture procedure was also significantly lower. Each step of the vascular procedure could be programmed and standardised in the simulation system. After the teacher's explanation and auxiliary training, it was easier for residents to develop standardised vascular skill habits and the correct procedural process. Finally, we evaluated the training effect with the cerebrovascular angiography procedure. Our results proved that the learning time of the angiography procedure and time of completed procedures were significantly lower in residents with ST; these results were consistent with previous reports.[14] This finding also confirmed that different simulation systems and teaching models could improve the effectiveness of clinical teaching and promote the understanding and proficiency of vascular practical performance.

Furthermore, the final effect of clinical training should be assessed by the practice procedure of residents. Our study confirmed that vascular ST could promote the practice knowledge of residents and promote the understanding of vascular surgical procedure. Most cases of vascular disease teaching need to be performed under radiation; thus, ST could avoid radiation damage to teachers, patients and residents. In this study, the results demonstrated that ST could decrease the fluo time and CAK of the procedure, which meant that ST could reduce the radiation damage of patients and residents, and radiation protection was an important teaching ethics component in vascular surgery practice. However, due to the limitations of our teaching funding and the time course, only selective residents underwent the ST for 2 weeks, which was different from what occurs in advanced vascular centres. Other reports have shown that ST for 8 weeks can improve the procedure skills of residents, contribute to patient safety and have a positive impact on the career planning and choice of vascular surgeons.[15] Therefore, the ST should be the basic course for residents. Therefore, the ST should be the basic course for residents, especially in precareer students and residents in rotation, meanwhile the ST should be well defined and step planned; otherwise, the ST may result in an impaired learning and worse performance in real clinical practice.[16]

### Limitations

There were several limitations. First, this study was not a prospective randomised study, and the conclusions of this study should be confirmed in the future. Second, when compared with western counties, the ST is not wildly used in China, thus the truly effect of the ST is not investigated deeply. Third, the clinical performance of the residents

should be evaluated via the real clinical procedure after the ST, this is the next step of our study.

## Conclusions

Our results confirmed that a vascular simulation system could improve the clinical skills of residents and reduce the radiation damage received by patients and residents in endovascular procedures.

**Acknowledgements** The language of this article was edited by American Journal Experts. Thanks to Professor Jian Yang from the clinical research centre of the First Affiliated Hospital of Xi'an Jiaotong University for providing statistical analysis support and proofread.

**Contributors** LY and YLi: data collection. LY, YLi, JL and YL: conception or design of the work; data analysis and interpretation; critical revision of the article; drafting of the article and final approval of the version to be published.

**Funding** This project was supported by the Natural Science Foundation of China (NO.8197 0365).

**Competing interests** None declared.

**Patient and public involvement** Patients and/or the public were not involved in the design, or conduct, or reporting, or dissemination plans of this research.

**Patient consent for publication** Not required.

**Ethics approval** This study was approved by the institutional review and ethics board of the First Affiliated Hospital of Xi'an Jiaotong University (XJTU1AF2014LSK-112), all of the patients provided written informed consent.

**Provenance and peer review** Not commissioned; externally peer reviewed.

**Data availability statement** No data are available. None

**ORCID iD**
Lin Yang http://orcid.org/0000-0001-5459-7373

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
