## [Reviewer comments · BMJ Open]

ARTICLE DETAILS

TITLE (PROVISIONAL)	The effect of vascular simulation training on practice performance in residents: a retrospective cohort study
AUTHORS	Yang, Lin; Li, Yanzi; Liu, Jianlin; Liu, Yamin

VERSION 1 – REVIEW

REVIEWER	Arman Azadi Ilam university of medical Science, Ilam, Iran.
REVIEW RETURNED	07-Feb-2020

GENERAL COMMENTS	The running head is not suitable for your title Abstract: Please mention the type of study at the first method section in the abstract in method how you divided the 95 residents into two groups? Also please mention the instruments you were used for data collection. Please select your keywords based on MeSH terms. Introduction Please clarify the necessity of conducting such a study using simulation. Please mention the gap of knowledge and how you want to fill this gap. Also there is no information about current knowledge in this topic. What is conventional teaching? Method Please mention the type of study. How you perform sampling and allocation to each ST and CT groups? Dose the ST group get simulation at first and then conventional teaching? Page 7 line 121. dose simulation system exist in the hospitals before the current study? page 8 line 138. You mentioned "the student scores" . Score for knowledge or practice? There is no information about ethical consideration Results: page 10 line 177. Theoretical score is equivalent to knowledge? Please be consistent in wording of concepts. How we could expect the simulation system have an effect on participants knowledge scores? Discussion Discussion is poor. The relevant studies dose not mentioned in a good manner. You have not developed simulation system. Such a
--

	system currently used in many countries. So please have a discussion about how to proceed using such system, what is the limitation of this system and so on. The limitation of study was not declared.
REVIEWER	Emil Petrusa, PhD Massachusetts General Hospital Department of Surgery and Learning Lab Harvard Medical School United States of America
REVIEW RETURNED	11-Mar-2020
GENERAL COMMENTS	This reviewer has confusion about "patients" comfort. Who are the "patients?" In brief section about "Patients and Public Involved" authors state that no actual patients were involved in the study, yet they report NRS (comfort) scores. Please explain who are the patients. Authors do not offer any "study limitations." Finally, Table 3 shows simulation group had significantly (LTP) less learning, but in the Study Procedures section authors state that ST students received ST first and then "completed last clinical training." This sounds like the ST group got BOTH ST and CT learning. Please clarify.

VERSION 1 – AUTHOR RESPONSE

Reviewer(s)' Comments to Author:

Reviewer: 1

Reviewer Name

Arman Azadi

Institution and Country

Ilam university of medical Science, Ilam, Iran.

Please state any competing interests or state 'None declared':

none

Please leave your comments for the authors below

The running head is not suitable for your title

A: Thank you very much for your advice, and we have corrected the running title of the manuscript in the revised version at page 1 and line 14.

Abstract:

Please mention the type of study at the first method section in the abstract in method how you divided the 95 residents into two groups?

Also please mention the instruments you were used for data collection.

Please select your keywords based on MeSH terms.

A: Thank you very much for your advice and we do apologize for these mistakes. We have modified the abstract and the keywords according to the guideline in revised manuscript at page 2-3 and line 24-50.

Introduction

Please clarify the necessity of conducting such a study using simulation. Please mention the gap of knowledge and how you want to fill this gap. Also there is no information about current knowledge in this topic. What is conventional teaching?

A: Thank you very much for your comments. We have corrected and re-arranged the introduction section in the revised manuscript at page 5-6 and line 94-111.

Method

Please mention the type of study.

A: Thank you very much for your advice. We have added the type of study in Abstract section and methods section in revised manuscript at page 2 line 26 and page 6 line 115-119.

How you perform sampling and allocation to each ST and CT groups?

A: Thank you very much for your comments. This study was a retrospective study, and the sampling and allocation were collected retrospectively. Because we need to arrange the use of simulation system after contact with education department according to the situation of the residents in the rotation, thus this study is difficult to conduct as a randomized study. The study design was added in the revised manuscript at abstract and method section at page 2 line 26 and page 6 line 115-119.

Dose the ST group get simulation at first and then conventional teaching?

A: Thank you very much for your comments. Yes, in this study and our clinical practice, the ST group get simulation training first and then conventional teaching.

Page 7 line 121. dose simulation system exist in the hospitals before the current study?

A: Thank you very much for your comment. Yes, the simulation system was bought by our hospital and could be used for every vascular-related specialty, therefore, each specialty only have limited time to use the simulation system.

page 8 line 138. You mentioned "the student scores" . Score for knowledge or practice?

A: Thank you very much for your comments and we do apologize for this confusion. In this study, the student scores were evaluated by the test, which including the information of knowledge and practice, this information was added in the revised manuscript at page 8 line 157-159.

There is no information about ethical consideration

A: Thank you very much for your comments and we do apologize for this error. We have corrected the information in the revised manuscript at page 6 line 123-125, page 14 line 305-306.

Results:

page 10 line 177. Theoretical score is equivalent to knowledge? Please be consistent in wording of concepts. How we could expect the simulation system have an effect on participants knowledge scores?

A: Thank you very much for your comments and we do apologize for this confusion. In the process of residents training, the theoretical score is not completely equal to the level of knowledge, however, the theoretical score is a way to evaluate the knowledge level of the residents. Meanwhile, residents could better understand theoretical knowledge and practical knowledge through simulation training , which also helps to improve the residents' theoretical scores. These information was added in the Discussion section in revised manuscript at page 11-12 line 242-245.

Discussion

Discussion is poor. The relevant studies dose not mentioned in a good manner. You have not developed simulation system. Such a system currently used in many countries. So please have a discussion about how to proceed using such system, what is the limitation of this system and so on.

A: Thank you very much for your comments. We have re-arranged the Discussion section according to related information in the revised manuscript page 11 line 230-232, page 11-12 line 240-248, page 13 line 267-269, 280-284.

The limitation of study was not declared.

A: Thank you very much for your advice and we do apologize for this mistake. The limitation section has been added in the revised manuscript at page 13-14 line 285-291.

Reviewer: 2

Reviewer Name

Emil Petrusa, PhD

Institution and Country

Massachusetts General Hospital

Department of Surgery and Learning Lab

Harvard Medical School

United States of America

Please state any competing interests or state 'None declared':

None declared

Please leave your comments for the authors below

This reviewer has confusion about "patients'" comfort. Who are the "patients?" In brief section about "Patients and Public Involved" authors state that no actual patients were involved in the study, yet they report NRS (comfort) scores. Please explain who are the patients.

A: Thank you very much for your comments and we do apologize for this confusion. In the section of "Patients and Public Involved", we state that patients and the public were not involved in the design or planning of the study. In the process of residents training, the patients here refer to the actual cases in the surgical practice, however, these patients did not involve in the design, conduction and evaluation of the study.

Authors do not offer any "study limitations."

A: Thank you very much for your comment and we do apologize for these mistake. We have modified the limitation section in the revised manuscript at page 13-14 and line 285-291.

Finally, Table 3 shows simulation group had significantly (LTP) less learning, but in the Study Procedures section authors state that ST students received ST first and then "completed last clinical training." This sounds like the ST group got BOTH ST and CT learning. Please clarify.

A: Thank you very much for your comments and we do apologize this confusion. In this study, the ST group received the both ST and CT learning, and the CT group only received the CT learning. We have re-arranged the definition of these parameters in the revised manuscript at page 8 line 171-176.

VERSION 2 – REVIEW

REVIEWER	Arman Azadi Ilam University of medical Sciences, Ilam, Iran
REVIEW RETURNED	13-Jun-2020

GENERAL COMMENTS	I appreciate the work of the authors. Most of issues raised in the previous round of review were resolved by authors. However, the study is not sufficiently described-insufficient background to support the study. There is insufficient sampling information and information about the design is insufficient. I consider that the proposed design is not
---

	of a retrospective cohort study. If you think this is retrospective cohort study please gives the rational in the text. Statistical analyses are too simplistic. Why you used T test and what is prerequisite for T test? There is insufficient information about study instruments and its reliability and validity. In the page 7 the authors mentioned they assessed theoretical knowledge. How the assessment was done and how you validate your instrument.
--	---

VERSION 2 – AUTHOR RESPONSE

Reviewer(s)' Comments to Author:

Reviewer: 1

Please leave your comments for the authors below

I appreciate the work of the authors. Most of issues raised in the previous round of review were resolved by authors. However, the study is not sufficiently described-insufficient background to support the study.

There is insufficient sampling information and information about the design is insufficient. I consider that the proposed design is not of a retrospective cohort study. If you think this is retrospective cohort study please gives the rational in the text. Statistical analyses are too simplistic. Why you used T test and what is prerequisite for T test? There is insufficient information about study instruments and its reliability and validity. In the page 7 the authors mentioned they assessed theoretical knowledge. How the assessment was done and how you validate your instrument.

A: Thank you very much for your comments and we do apologize for the confusion of the manuscript, we have modified the manuscript according to the reviewer’s comments in the revised version.

Due to the late entry of simulation systems in China, in our hospital, after a period of application, we analyzed the data retrospectively and the results confirmed the effectiveness of the simulation training. Thereafter, we have written this retrospective research paper. The modified in formation were added in the revised manuscript at page 2 line 27 and page 5-6 line 109-111.

Furthermore, in most studies comparing two samples, t-test and chi-square test are the most commonly used test methods. Therefore, we also use these methods in our research. We are very appreciating your professional opinions and thank you for your kind help to keep improving the details of the paper. Meanwhile, the statistical analysis of the manuscript is supported and proofread by statistical analysis experts in our university. We added the words of thanks at the acknowledgment in the revised manuscript at page 14 line 304-306.

In our study, we have evaluated the effectiveness of the simulation system. The reliability and effectiveness of this tool has been supported by several documents, such as Ref. 3-6 and 13-15. Our research has further confirmed the effectiveness and universal applicability of the simulation system, and it is worthy of further popularization and application.

The theoretical knowledge test is formulated according to the standardized question bank and the simulation system question bank, which helps avoids the bias due to the teacher or students' selection. The modified words were added in the revised manuscript at page 8 line 158-160.